# *Xanthium strumarium* Fruit Extract Inhibits ATG4B and Diminishes the Proliferation and Metastatic Characteristics of Colorectal Cancer Cells

**DOI:** 10.3390/toxins11060313

**Published:** 2019-06-02

**Authors:** Hsueh-Wei Chang, Pei-Feng Liu, Wei-Lun Tsai, Wan-Hsiang Hu, Yu-Chang Hu, Hsiu-Chen Yang, Wei-Yu Lin, Jing-Ru Weng, Chih-Wen Shu

**Affiliations:** 1Department of Biomedical Science and Environmental Biology, Kaohsiung Medical University, Kaohsiung 80708, Taiwan; changhw@kmu.edu.tw; 2Department of Medical Research, Kaohsiung Medical University Hospital, Kaohsiung 80708, Taiwan; 3Drug Development and Value Creation Research Center, Kaohsiung Medical University, Kaohsiung 80708, Taiwan; 4Department of Medical Education and Research, Kaohsiung Veterans General Hospital, Kaohsiung 81362, Taiwan; pfliu@vghks.gov.tw (P.-F.L.); mini16610@gmail.com (H.-C.Y.); 5Division of Gastroenterology, Department of Internal Medicine, Kaohsiung Veterans General Hospital, Kaohsiung 813, Taiwan; tsaiwl@yahoo.com.tw; 6Department of Colorectal Surgery, Kaohsiung Chang Gung Memorial Hospital and Chang Gung University College of Medicine, Kaohsiung 83341, Taiwan; gary.hu0805@msa.hinet.net; 7Department of Radiation Oncology, Kaohsiung Veterans General Hospital, Kaohsiung 81362, Taiwan; ychu81@gmail.com; 8Department of Pharmacy, Kinmen Hospital, Kinmen 89142, Taiwan; u8557006@gmail.com; 9Department of Marine Biotechnology and Resources, National Sun Yat-sen University, Kaohsiung 80424, Taiwan; 10School of Medicine for International Students, I-Shou University, Kaohsiung 82445, Taiwan

**Keywords:** medicinal plant, autophagy, ATG4B, colorectal cancer

## Abstract

Autophagy is an evolutionarily conserved pathway to degrade damaged proteins and organelles for subsequent recycling in cells during times of nutrient deprivation. This process plays an important role in tumor development and progression, allowing cancer cells to survive in nutrient-poor environments. The plant kingdom provides a powerful source for new drug development to treat cancer. Several plant extracts induce autophagy in cancer cells. However, little is known about the role of plant extracts in autophagy inhibition, particularly autophagy-related (ATG) proteins. In this study, we employed S-tagged gamma-aminobutyric acid receptor associated protein like 2 (GABARAPL2) as a reporter to screen 48 plant extracts for their effects on the activity of autophagy protease ATG4B. *Xanthium strumarium* and *Tribulus terrestris* fruit extracts were validated as potential ATG4B inhibitors by another reporter substrate MAP1LC3B-PLA_2_. The inhibitory effects of the extracts on cellular ATG4B and autophagic flux were further confirmed. Moreover, the plant extracts significantly reduced colorectal cancer cell viability and sensitized cancer cells to starvation conditions. The fruit extract of *X. strumarium* consistently diminished cancer cell migration and invasion. Taken together, the results showed that the fruit of *X. strumarium* may have an active ingredient to inhibit ATG4B and suppress the proliferation and metastatic characteristics of colorectal cancer cells.

## 1. Introduction

Autophagy is a self-eating pathway in eukaryotic cells that recycles proteins and organelles to generate substrates for new synthesis in cells during times of nutrient deprivation or stresses. This evolutionarily conserved pathway is highly associated with many diseases, particularly cancer progression [1]. Autophagy eliminates Sequestosome-1(SQSTM1)-mediated spontaneous tumorigenesis and functions as a tumor suppressor during cancer development [2]. By contrast, autophagy is elevated to protect cancer cells under stressed conditions, including hypoxia and chemotherapy [3,4]. Autophagy inhibitors or silencing essential autophagy genes increases the chemosensitivity of tumor cells in vitro and *in vivo* [3,5]. Based on autophagy as an oncogenic pathway, the autophagy inhibitor hydroxy chloroquine (HCQ), also known as an anti-malaria drug, has been tested in more than 30 clinical trials for cancer therapy. It has been validated to have antitumor effects in a certain subset of cancer patients, such as those with glioma and colorectal cancer [6]. These results indicate that the autophagy inhibitor may improve cancer therapy in certain types of cancer. 

Approximately 38 autophagy-related (ATG) genes are primarily involved in the core machinery of autophagy in mammalian cells. ATG4 is the cysteine protease required to activate microtubule-associated light chain 3 precursor (proMAP1LC3) for further conjugation with phospholipid, while it is also involved in the deconjugation of membraned-bound MAP1LC3 (MAP1LC3-II) from autophagosomes for recycling [7]. There are four ATG4 members in mammalian cells, ATG4A, ATG4B, ATG4C and ATG4D. ATG4B has the highest and broadest proteolytic activity on all substrates, such as MAP1LC3 and gamma-aminobutyric acid receptor associated protein like 2 (GABARAPL2) [8,9]. ATG4B expression in tumor tissues is much higher than that in adjacent normal cells of colorectal cancer patients [10]. Likewise, upregulated ATG4B is associated with drug resistance in CD34^+^ chronic myeloid leukemia (CML) patients [11]. Silencing ATG4B significantly suppresses cancer cell growth [12] and synergizes the killing effects of trastuzumab in HER2-positive breast cancer cells [13]. Ectopic expression of the dominant-negative mutant ATG4B^C74A^ diminishes cell proliferation in hepatocellular cell carcinoma [14]. These results suggest that ATG4B might be a potential drug target for cancer therapy. Small-molecule inhibitors of ATG4B have been developed based on biochemical screening or in silico screening [15,16,17]. However, the inhibitory effects of these small molecules in cancer cells are unclear. 

The plant kingdom has provided a huge resource for medicinal use as herbal medicines in various formulations, such as infusions, syrups, and ointments. Traditional use of medicinal plants includes knowledge, skills, and practices to prevent and treat numerous diseases [18], including cancer [19]. Thus far, approximately 25% of all prescriptions contain at least one active ingredient obtained from medicinal plants [20]. Some of the bioactive ingredients derived from plants are used as candidates for drug development [21]. Several plant extracts have been reported as comprising autophagy inducers, such as resveratrol [22], curcumin [23], and D-limonene [24]. Nevertheless, little is known about the role of plant extracts in autophagy inhibition, particular regarding certain ATG proteins. In this study, we exploited edible medicinal plants to screen for potential ATG4B inhibitors and determine their effects in cancer. Our results showed that extracts from *Xanthium strumarium* and *Tribulus terrestris* effectively blocked ATG4B activity in vitro. The extracts consistently inhibited colorectal cancer cell viability and sensitized cancer cells to starvation, which is an autophagy-inducing condition. Moreover, the extracts from *X. strumarium* diminished the migration and invasion of colorectal cancer cells, suggesting the ingredients of *X. strumarium* may inhibit ATG4B and have anti-cancer effects in colorectal cancer cells.

## 2. Results

### 2.1. Screening Plant Extracts for Potential ATG4B Inhibitors 

All plants used in this study were typical Formosan plants from southern Taiwan, which were collected and identified by one of the co-authors, Dr. Wei-Yu Lin in Ping Tung County, Taiwan. It was possible that different parts of each plant would demonstrate different anti-ATG4B activity, which led us to collect the different parts. Also, comparing different parts of the same plant would indicate which of these parts exhibits the highest anti-ATG4B activity. This is the same reason why we used different kinds of solvents for the extraction. The plants were ground and extracted with the indicated solvent at room temperature for one week. After three extractions, the extracts were concentrated under vacuum for two weeks to remove the solvent. Thus, the aim of this study was to screen different Formosan plants extracts for potential inhibitors against ATG4B and/or autophagy, as a starting point for cancer therapy. 

To explore whether any Formosan plant extracts that we had could inhibit ATG4B activity, we collected 23 plants and extracted different parts of the plants, including the stem, leaves, fruit, root, and heartwood (Table 1). Crude extracts from some of these parts were obtained by various solvents, such as acetone, methanol (MeOH), and chloroform (CHCl_3_), to harvest 48 extracts, as shown in Table 1. To determine the inhibitory effects of extracts on ATG4B proteolytic activity, the chimeric protein GABARAPL2, with an N-terminal C-myc tag and a C-terminal S tag, was generated and purified. ATG4B can cleave the C-terminus of the GABARAPL2 chimeric protein to release the short peptide of the S tag, as shown in the schematic diagram in Figure 1A. Recombinant ATG4B and GABARAPL2 were mixed with 10-fold titrated medicinal plant extracts, and cleavage activity was examined by immunoblotting using antibodies against the S tag or C-myc tag. The released S-tag could not be detected in the immunoblotting gel, due to the small molecular weight, while the retained S-tag and upper band of the C-myc tag (full-length GABARAPL2) represented the inhibitory effects of ATG4B on GABARAPL2, as the control (+, Figure 1B). Compared to the controls, the remaining S-tag and C-myc tag levels were quantified to determine the inhibitory effects of medicinal plants extracts on ATG4B proteolytic activity toward GABARAPL2 (Figure 1C,D). Among these extracts, *Xanthium strumarium* and *Tribulus terrestris* extracts from different solvents (number 33–36) showed consistent results in the retained S-tag and C-myc tag, suggesting the extracts may contain active ingredients to inhibit ATG4. Two of the plant extracts (21 and 26) were randomly tested in different vials and blots to determine if the assay was consistent. On the other hand, a high dose of some plant extracts decreased the recombinant ATG4B level but had no or little effects on GABARAPL2 cleavage, likely due to aggregation or degradation of ATG4B, caused by a high dose of the compounds. Thus, we set these extracts aside, and the most promising extracts from the fruit of *X. strumarium* (XS-(fr)-M and XS-(fr)-C) and *T. terrestris* (TT-(fr)-A and TT-(fr)-M) were further tested using another biochemical assay with the MAP1LC3B-PLA_2_ reporter protein. 

Briefly, MAP1LC3B, another substrate of ATG4B, was used to replace the prodomain of PLA_2_ to keep it in the inactive form. Once MAP1LC3B was cleaved by recombinant ATG4B, PLA_2_ was activated and could be assayed by the fluorescent substrate NBD-C6-HPC, as previously reported [16]. Consistent with the cleavage results in Figure 1, these two medicinal plant extracts suppressed the fluorescent signal in the LC3B-PLA_2_ reporter assay (Figure 2), supporting the notion that the fruit extracts from *X. strumarium* and *T. terrestris* inhibited ATG4B proteolytic activity. 

### 2.2. Validation of the Effect of the Plant Extracts on Cellular ATG4B Activity and Autophagic Activity

Because the fruit extracts inhibited essential autophagy protease ATG4 in vitro, we further inspected whether the extracts inhibited cellular ATG4B activity and autophagic flux. The cells were transfected with ATG4-cleavable reporter luciferase, in which luciferase activity can be restored when cellular ATG4B is inhibited [5]. The transfected cells were then treated with the fruit extracts, including XS-(fr)-M, XS-(fr)-C, TT-(fr)-A, and TT-(fr)-M (Figure 3A). These four extracts increased luciferase activity compared with cells without treatment (Figure 3A). Moreover, human colorectal cancer cells were treated with fruit extracts, as mentioned above, in the presence or absence of the autophagy inhibitor BafA1, followed by measurement of MAP1LC3B flux to reflect autophagy activity (Figure 3B). A high dose (50 μg/mL) of fruit extracts from *Xanthium strumarium* and *Tribulus terrestris* increased the MAP1LC3-II protein level. The net MAP1LC3-II level in cells between extracts treated alone or in combination with BafA1 was decreased compared with the control cells (Figure 3C), particularly in XS-(fr)-C- and TT-(fr)-M-treated cells. 

### 2.3. Effects of Plant Extracts on Colorectal Cancer Cells

Autophagy and ATG4 are involved in cell proliferation and survival in cells during autophagy inducing conditions, particularly in starvation [4,5]. To inspect whether the ATG4-inhibitory extracts can inhibit cell proliferation and sensitize cancer cells to starvation, human colorectal HCT116 or DLD-1 cells were further treated with extracts from *X. strumarium* (XS-(fr)-M and XS-(fr)-C) and *T. terrestris* (TT-(fr)-A and TT-(fr)-M) (Figure 4). The extracts significantly decreased the cell viability of both HCT116 and DLD-1 cells. Interestingly, fruit extracted from certain solvents (XS-(fr)-C and TT-(fr)-M) synergized the inhibitory effects on cancer cells under starved conditions (EBSS), whereas XS-(fr)-M and TT-(fr)-A had no synergistic effects with the autophagy inducer in cancer cells (Figure 4). These results implied that XS-(fr)-C and TT-(fr)-M might be more suitable for further experiments. 

In addition to the role of autophagy in cell proliferation and survival, autophagy plays a crucial role in cancer metastasis [25]. Cancer cell migration and invasion are typical features involved in cancer metastasis. Since 10 or 50 μg/mL of XS-(fr)-C or TT-(fr)-M had no or little effects on cell viability (Appendix A), human colorectal cancer HCT116 and DLD-1 cells were treated with 10 or 50 μg/mL XS-(fr)-C or TT-(fr)-M for the cell migration assay (Figure 5). XS-(fr)-C significantly reduced the cell migration of HCT116 and DLD-1 cells (Figure 5A,B), while TT-(fr)-M had little or minor effects on the reduced migration in HCT116 cells (Figure 5C). By contrast, 50 μg/mL of TT-(fr)-M slightly increased the cell migration in DLD-1 cells (Figure 5D). The colorectal cancer cells were further treated with XS-(fr)-C or TT-(fr)-M for the cancer cell invasion assay (Figure 6). Both XS-(fr)-C or TT-(fr)-M significantly decreased the invaded cells, particularly in HCT116 cells. In addition, compared to two-dimensional culture, three-dimensional tumor cell culture is relatively more capable of reproducing complicated microenvironments *in vivo*, such as low nutrients and oxygen in central part of tumors, which induces autophagy. Therefore, to precisely inspect the effects of plant extracts in cancer cells, HCT116 cells were cultured for spheroid formation and treated with XS-(fr)-C or TT-(fr)-M (Figure 6E,F). We found XS-(fr)-C or TT-(fr)-M significantly increased dead cells in tumor spheres. Taken together, these results indicate that XS-(fr)-C may be more potent than TT-(fr)-M for inhibiting cancer malignant characteristics. 

## 3. Discussion

ATG4B is highly expressed in tumor tissues, compared to the expression in normal cells, in different types of cancer, such as colorectal cancer [10], CML, and breast cancer [13]. Inactivation or silencing of ATG4B also decreased the cell viability in various cancer cells, including hepatoma cells, breast cancer cells, glioma, and colorectal cancer cells [5,12]. Our current results showed that *X. strumarium* and *T. terrestris* extracts could inhibit ATG4B and reduce the cell viability of colorectal cancer cells. The suppressive effects of these extracts on other types of cancer cells might be worthy of elucidation. Moreover, ATG4B is one of four ATG4 members in mammalian cells. ATG4A shows high homology with ATG4B and has proteolytic activity on GABARAPL2 [8,9], whereas ATG4C and ATG4D are inactive forms before their cleavage by caspase-3 [26]. The effects of *X. strumarium* and *T. terrestris* extracts on the other ATG4 members in cells during normal or apoptotic conditions would need further work for verification. 

Although autophagy may function as a suppressor in tumorigenesis, by reducing SQSTM1-mediated inflammation and proliferation pathways [27], an increasing number of reports has shown that autophagy contributes to cancer malignancy and stemness: (I) high throughput screening using a human shRNA library revealed that ATG4A is required for tumor spheroid formation [28]; (II) BECN1, a mammalian homolog of ATG6, facilitated cancer stem cells (CSCs) survival and tumor formation in a xenografted mouse model [29]; III) autophagy deficiency decreases osteosarcoma CSCs and results in elevated chemosensitivity [30,31], suggesting that autophagy may not only promote cancer cell survival but also facilitate cancer stemness. Because our present results showed that the fruit extracts from *X. strumarium* and *T. terrestris* can block autophagic flux, these results suggest the extracts might inhibit tumor progression in certain types of cancer or conditions. On the other hand, the *T. terrestris* extract reduced proliferation but enhanced migration. A previous study has shown that YB-1 translation represses pro-growth genes and activates epithelial-to-mesenchymal transition (EMT)-related genes [32], raising the possibility that the *T. terrestris* extract might activate YB-1 in colorectal cancer cells. Additionally, migration and invasion are common features in cancer metastasis. Elevated migration is usually positively correlated with increased invasion. Nevertheless, the migration phenotype is sometimes uncoupled with invasion, likely through the activation of EGFR and EMT [33]. In line with a previous study, *T. terrestris* extract increased cancer cell migration but inhibited the invasiveness of cancer cells. The molecular mechanisms underlying the inconsistent effects of *T. terrestris* extract on proliferation, migration, and invasion require more work for clarification.

Autophagy plays a cytoprotective role in plant extract-mediated cancer cell inhibition. Resveratrol, an abundant polyphenolic phytoalexin in grapes, peanuts, and berries, activates autophagy by reducing the Akt/mTOR pathway [34]. The autophagy inhibitor 3-methyladenine (3-MA) increases resveratrol-induced apoptosis in non-small cell lung cancer cells [34]. Similarly, curcumin was found to modulate the Akt/mTOR pathway, leading to the reduction of cancer cell proliferation [35,36]. Inhibiting autophagy with 3-MA enhances curcumin-induced apoptosis in gastric cancer cells [35]. Moreover, salidroside, a phenylpropanoid glycoside from *Rhodiola rosea*, induces AMPK phosphorylation to elevate autophagy in colorectal cancer cells [37]. The AMPK antagonist compound C or autophagy inhibitor 3-MA synergizes cancer cells to chemotherapeutic agents, including oxaliplatin, 5-fluorouracil, and doxorubicin [37]. Our present study showed that *X. strumarium* fruit extracts inhibit the essential autophagy protease ATG4B to block autophagy and cell viability. These results suggest that the combination of *X. strumarium* fruit extracts with autophagy-inducing compounds, as mentioned above, might provide a natural method for cancer therapy. 

## 4. Conclusions

Current evidence has implicated the importance of autophagy in the survival of cancer cells, in the context of nutrient deprivation and other stressful circumstances. ATG4B is a cytosolic cysteine protease required for autophagy machinery, and it is elevated in several types of cancer, particularly colorectal cancer. Recent studies have shown that the knockdown of ATG4B can suppress cancer cells [12]. On the other hand, medicinal plants have found their potential in autophagy modulation and tumor suppression [34,35]. However, the effects of plant extracts on ATG proteins remain unclear. In this study, we reported the following findings. First, we screened 48 plant extracts using the biochemical reporter assay and identified that the fruit extracts from *X. strumarium* and *T. terrestris* inhibited ATG4 proteolytic activity. Second, the extracts reduced cellular ATG4B activity and autophagic flux. Third, *X. strumarium* and *T. terrestris* extracted from certain solvents decreased the cell viability and had synergistic effects on cancer cell suppression during starvation. Fourth, *X. strumarium* fruit extract reduced colorectal cancer cell migration and invasion. Our study suggested that *X. strumarium* fruit might contain active ingredients to inhibit colorectal cancer cells through ATG4B suppression. 

## 5. Materials and Methods 

### 5.1. Plasmid Construction

The human ATG4B gene and the genes of the substrate proteins GABARAPL2 and MAP1LC3B were cloned into the bacterial expression vector pETDuet-1 (Novogen, 71146-3) and in-frame with an N-terminal His tag and a C-terminal S-tag. GABARAPL2 and MAP1LC3B also contained a C-myc tag in the N-terminus, as mentioned previously [9]. The human GABARAPL2 or MAP1LC3B and PLA_2_ group X G10 (amino acid 43–165) were amplified to make a fusion protein, GABARAPL2-PLA_2_ or MAP1LC3B-PLA_2_ in pETDuet-1 [10,16]. The ATG4-cleavable reporter plasmid was generated using the MAP1LC3B-luciferase chimeric gene, as described previously [5]. The MAP1LC3B (G120A) mutant was used as a non-cleavable reporter plasmid to normalize the luciferase activity and reflect ATG4 activity. 

### 5.2. Plant Extract Isolation

The 23 Formosan plants were collected from Pingtung, in the southern part of Taiwan. In the 48 active samples, only *Phytolacca Americana* (No. 7), *Asparagus cochinchinensis* (No.40), *Sida acuta* (No.46), and *Scrophularia ningpoensis* (No.47) contained an underground part. The plants were then subjected to tap water rinse cycles until soil particles could not be observed, and having been classified, dried and chopped very carefully. The contamination from soil bacterium should be eliminated as much as possible. The different parts of the plants were then extracted using different solvents, as described previously [38]. Briefly, various plants were ground, extracted with 5 mL of the indicated solvents, such as methanol, chloroform, and acetone, for 1 week. The extracts were further concentrated by vacuum for 2 weeks. These species were identified by Dr. Wei-Yu Lin, Department of Pharmacy, Kinmen Hospital, Kinmen 89142, Taiwan.

### 5.3. Protein Purification and Activity Assay 

The protein purification and activity assay were slightly modified from that in a previous report [9,16]. Briefly, the plasmids encoding ATG4B and GABARAPL2 were transformed into *E. coli* BL21 (DE3; Invitrogen; C6010-03, Carlsbad, CA, USA) and induced by 0.5 mM Isopropyl β-D-1-thiogalactopyranoside (IPTG) at room temperature for 4 h. All the recombinant proteins contained His-tag, which was purified by Ni-NTA-agarose (Qiagen, 30250, Germantown, MD, USA) according to the manufacturer’s instructions. For His-tagged GABARAPL2-PLA_2_ or MAP1LC3B-PLA_2_ protein purification, the inclusion bodies were denatured with 6 M guanidine hydrochloride, and the proteins were refolded in a column with an 8–0 M urea gradient, as reported previously [16]. Recombinant ATG4B was mixed with either S-tag or the PLA_2_ fusion substrate in the reaction buffer containing 20 mM Tris-HCl, pH 8.0, 100 mM NaCl, 2 mM CaCl_2_, and 1 mM DTT to determine proteolytic activity using immunoblotting or the PLA_2_ fluorescence assay [5]. 

### 5.4. Autophagic Flux and Immunoblotting 

For MAP1LC3B flux, HCT116 and DLD-1 cells (BCRC, Hsinchu, Taiwan) were treated with compounds for 6 h in the presence or absence of the autophagy inhibitor bafilomycin A1 (BafA1). The cells were lysed with a lysis buffer (1% NP40, 50 mM Tris Cl pH 7.5, 150 mM NaCl, 0.25% sodium deoxycholate, 1% SDS, protease inhibitor cocktail). Equal amounts of proteins were separated by SDS-PAGE and were transferred to nitrocellulose membranes. The membranes were blocked with skim milk and then were incubated with the primary antibodies against human ATG4B (A2981), MAP1LC3 (L7543), and ACTB (A5441) (purchased from Sigma-Aldrich) or TUBA (GTX628802, GeneTex, Irvine, CA, USA). The protein levels were then detected using the IRDye^TM^ 800 or Alexa 680 fluorescently labeled secondary antibody and the Odyssey infrared imaging system (LI-COR, Lincoln, NE, USA). 

### 5.5. Cell Viability Assay

The cell viability of colorectal cancer cells treated with plant extracts was determined using the CellTiter-Glo luminescent cell viability assay (Promega, Madison, WI, USA). Briefly, 0.5 × 10^5^ cells were cultured in 96-well plates for 24 h and then were treated with plant extracts for 24 h. The treated cells were further mixed with the CellTiter-Glo reagent to lyse the cells for 10 min. The luminescence signal was measured in a luminometer to reflect the cellular ATP levels and cell viability. For tumor sphere formation, the cells were cultured in 24-well NanoCulture plates (SCIVAX Corporation, Kawasaki, Japan) for seven days to form spheres. The tumor spheres were treated with plant extracts for 48 h, and then stained with Calcein AM (1 μM) and Ethidium homodimer-1 (EthD-1, 2 μM) (LIVE/DEAD^®^ Viability/Cytotoxicity Kit, Thermo Fisher Scientific, Waltham, MA, USA), as described previously [5]. The live (green) and dead (red) spheres were observed under fluorescence microscopy and quantitated by a Fluoroskan Ascent FL reader (Thermo Fisher Scientific). 

### 5.6. Wound-Healing Assay 

For the wound-healing assay, dishes were attached with IBIDI culture inserts (IBIDI, Inc., Planegg, Germany). HCT116 or DLD-1 cells in FBS-free DMEM medium (Invitrogen-Gibco, Carlsbad, CA, USA) were seeded at a density of 1.5 × 10^5^ cells/mL in the insert and were cultured overnight at 37 °C with 5% CO_2_. The confluent monolayers of cells were rinsed with PBS to remove residual cell debris at least twice and then were switched to DMEM with 0.5% serum for 12 h. The wounds were imaged and quantified by ImageJ software.

### 5.7. Cell Invasion Assays

Transwell invasion assays were performed using 8 μm pore inserts (Greiner Bio-One, Stroud, UK). The upper side of the filter was covered with 0.5% Matrigel (Collaborative Research, Boston, MA, USA). HCT116 or DLD-1 (1 × 10^5^) cells in 300 µL of DMEM containing 1% FBS were seeded into the top chamber. Complete medium was added to the bottom wells to stimulate invasion. The invasive cells on the underside of the membrane were fixed with 4% formaldehyde and stained with 0.1% crystal violet. The number of cells that invaded across the membrane was counted using a microscope and a ×200 objective, and was quantified by Prism 5.0.

### 5.8. Statistical Analysis

All data were expressed as the mean ± SEM from at least three individual experiments. The statistical analysis was performed by one-way Analysis of variance (ANOVA) with Tukey’s post hoc test. *p* values less than 0.05 were considered significant (* *p* < 0.05, ** *p* < 0.01, *** *p* < 0.001).

## Figures and Tables

**Figure 1 toxins-11-00313-f001:**
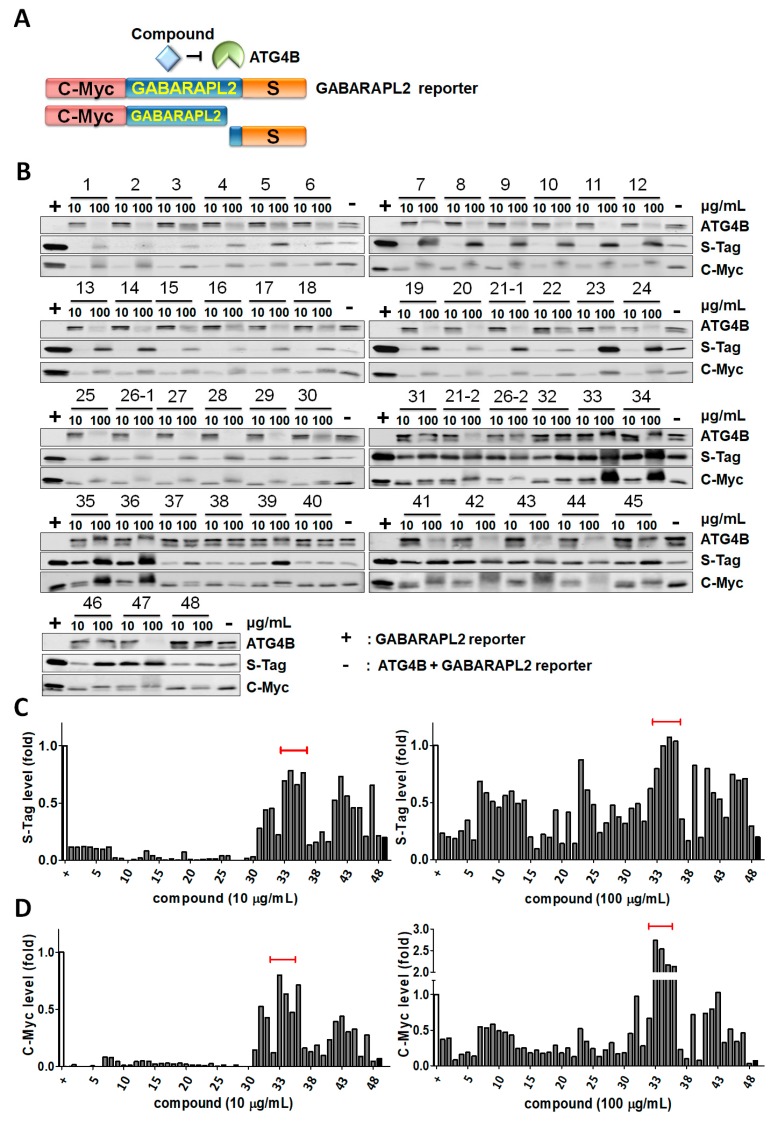
Screening for ATG4B inhibitors with medicinal plants using ATG4B cleavage assays. (**A**) Schematic diagram for the biochemical reaction to measure ATG4B proteolytic activity. Briefly, the substrate GABARAPL2 fused with the N-terminal C-myc tag and C-terminal S-tag. If extracts inhibited ATG4B, both full-length C-myc-tagged GABARAP2 (upper band) and a short peptide of S-tag can be detected by immunoblotting. (**B**) Recombinant ATG4B (5 nM) was mixed with the GABARAPL2 fusion protein (1 μM) in a reaction buffer containing 50 mM Tris-base, pH 8.0, 150 mM NaCl, and 1 mM DTT with 10-fold diluted formosan plant extracts (10 or 100 μg/mL). The retained S-tag and full-length C-myc tag (upper band) were determined by immunoblotting. The reactions without (+) and with (−) recombinant ATG4B were used as positive and negative controls of inhibition, respectively. (**C**) The protein levels of S-tag and (**D**) full-length C-myc-tagged GABARAPL2 were quantitated by ImageJ and normalized by positive control (+) as 100% inhibition of ATG4B in each blot.

**Figure 2 toxins-11-00313-f002:**
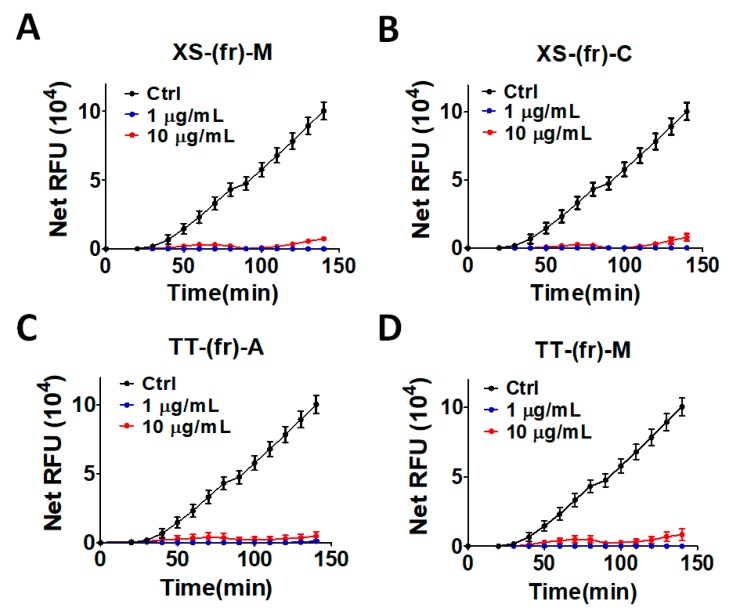
Hits validation of ATG4B inhibitors using the LC3B-PLA_2_ biochemical assay. Four hits obtained from the ATG4B cleavage assay—(**A**) XS-(fr)-M, (**B**) XS-(fr)-C, (**C**) TT-(fr)-A and (**D**) TT-(fr)-M—were reconfirmed using the MAP1LC3B-PLA_2_ biochemical assay. Medicinal plant extracts were mixed with recombinant ATG4B (0.5 nM) in the presence of 500 nM of the reporter MAP1LC3B-PLA_2_, another substrate for ATG4B, in a reaction buffer to determine the inhibitory effects of extracts on ATG4B activity. The fluorescence intensity was repeatedly measured for 150 min with excitation and emission wavelength of 485 and 530, respectively. The results are expressed as the mean ± SEM from three individual experiments.

**Figure 3 toxins-11-00313-f003:**
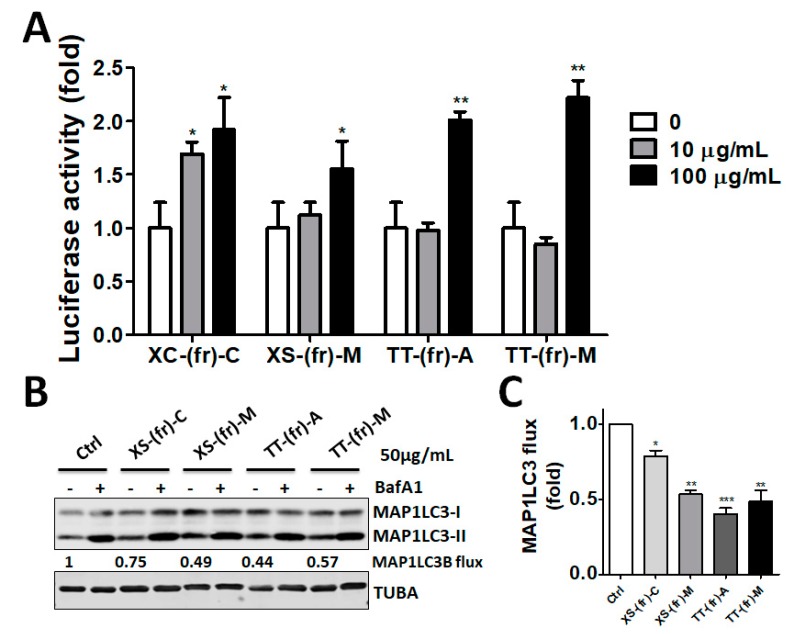
Effects of medicinal plants on cellular ATG4B and autophagic flux in human cells. (**A**) HEK293T cells were transfected with ATG4-cleavable luciferase and then were treated with plant extracts, including XS-(fr)-M, XS-(fr)-C, TT-(fr)-A, and TT-(fr)-M, for 6 h. The cells were mixed with One-Glo to measure luciferase activity. Increased luciferase activity indicated ATG4B inhibition, as described in the method. (**B**) To measure autophagic flux, HCT116 cells were treated with medicinal plant extracts in a cultured medium in the presence or absence of the autophagy inhibitor bafilomycin A1(B, 100 nM) for 3 h. The cells were then harvested for immunoblotting, using antibodies against MAP1LC3B and TUBA to determine MAP1LC3B-II accumulation to reflect autophagic flux. (**C**) The quantitative results for panel B were expressed as the means ± SEM from three independent experiments. The results are expressed as the mean ± SEM from 3 individual experiments. * *p* < 0.05; ** *p* < 0.01; *** *p* < 0.001 vs. cells treated with DMSO (0 or Control).

**Figure 4 toxins-11-00313-f004:**
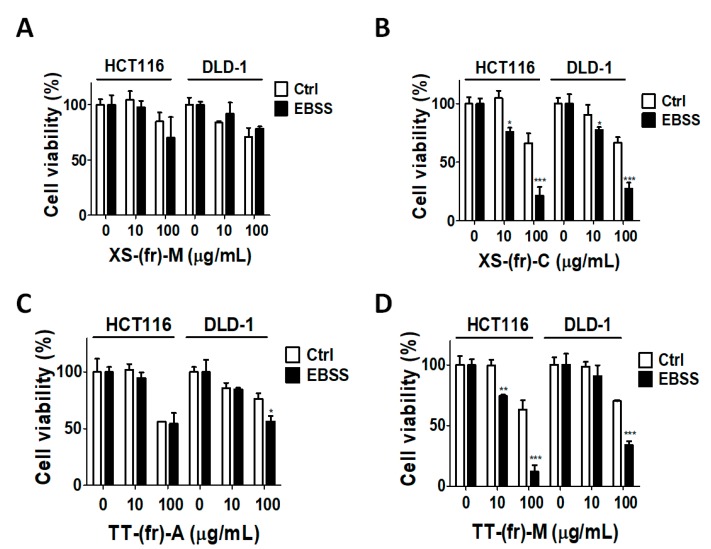
Effects of compounds on cell viability in colorectal cancer cells. Human colorectal cancer HCT116 and DLD-1 cells were cultured and treated with 10-fold diluted (10 or 100 μg/mL) extracts (**A**) XS-(fr)-M, (**B**) XS-(fr)-C, (**C**) TT-(fr)-A, (**D**) TT-(fr)-M in cultured medium (DMEM) or autophagy-inducing medium Earle’s Balanced Salt Solution (EBSS) for 24 h. The CellTiter-Glo reagent was added to cells, followed by measurement of the cellular ATP level to reflect cell viability. The results are expressed as the mean ± SEM from three individual experiments. * *p* < 0.05; ** *p* < 0.01; *** *p* < 0.001 for cells in starvation medium EBSS vs. cells with the same treatment in regular medium (Control).

**Figure 5 toxins-11-00313-f005:**
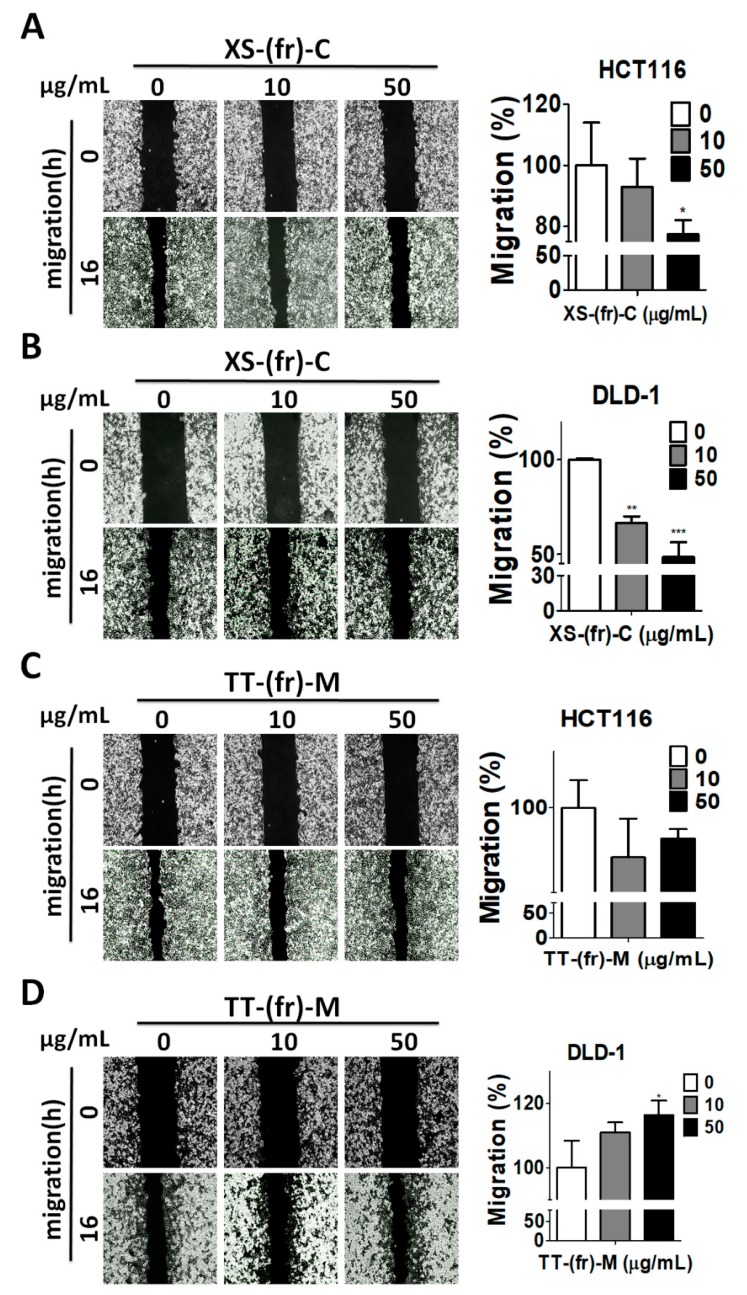
Effects of plant extracts on the migration of human colorectal cancer cells. Human colorectal cancer HCT116 (**A**,**C**) or DLD-1 (**B**,**D**) cells were seeded in culture inserts overnight and then were treated with plant extracts XS-fr-C (**A**,**B**) or TT-fr-M (**C**,**D**) for 16 h. The wounds were imaged and quantified by ImageJ software. The effects of plant extracts on cell migration were analyzed by Prism 5.0. The results are expressed as the mean ± SEM from three individual experiments. * *p* < 0.05; ** *p* < 0.01; *** *p* < 0.001 vs. cells treated with DMSO (0).

**Figure 6 toxins-11-00313-f006:**
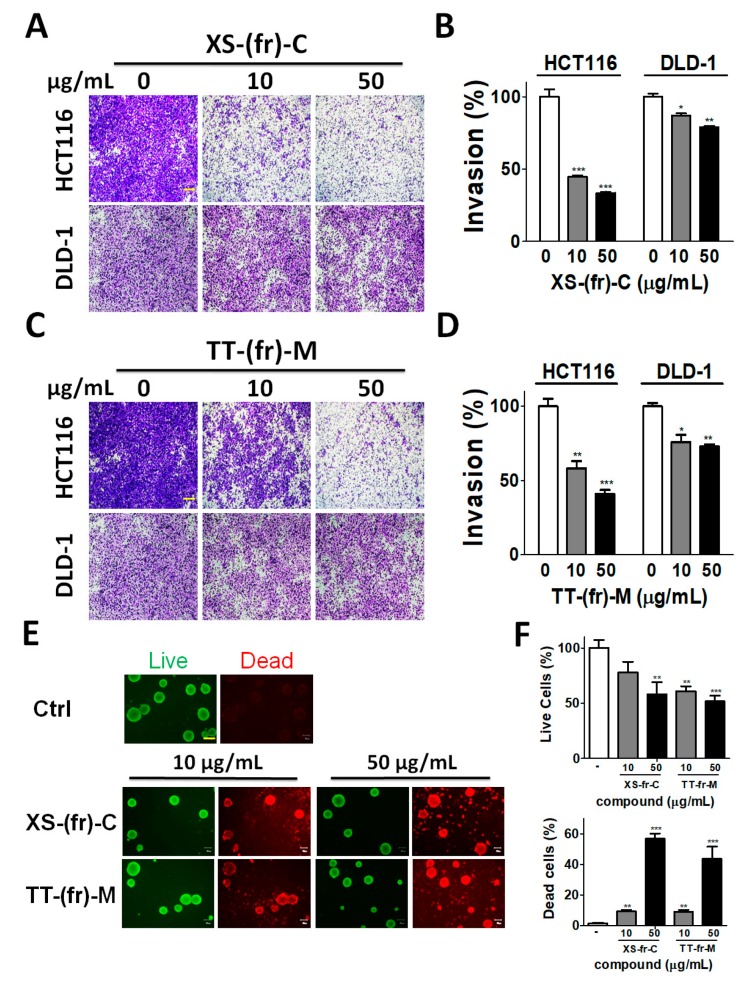
Effects of plant extracts on cell invasion in colorectal cancer cells. Human colorectal cancer HCT116 (**A**) or DLD-1 (**C**) cells were seeded on the top chamber of Matrigel-coated filters with low-FBS medium in the presence or absence (-) of plant extracts (**A**) XS-fr-C or (**C**) TT-fr-M. After invasion for 24 h, the filters were removed for fixation and stained with crystal violet. Scale Bars: 50 µm. The number of invaded cells was imaged with a microscope. A representative image is shown, and quantitative results (**B**,**D**) were expressed as the means ± SEM from three independent experiments. The results are expressed as the mean ± SEM from three individual experiments. * *p* < 0.05; ** *p* < 0.01; *** *p* < 0.001 vs. cells treated with DMSO (0). (**E**) HCT116 cells were cultured on NanoCulture plates for tumor sphere formation and treated with XS-fr-C or TT-fr-M for 48 h. The viable (green) and dead (red) cells were stained by a LIVE/DEAD staining kit. The representative images are shown. (**F**) The viable and dead cells were quantitated by a florescent reader. Scale Bars: 100 µm.

**Table 1 toxins-11-00313-t001:** Plant species tested for potential ATG4B inhibitors.

No.	Botanical Name	Extract	Family	Part of Plant
1	*Saurauia tristyla* var. *oldhamii*	Acetone	Rubiaceae	Leaves
2		CHCl_3_		Leaves
3		MeOH		Leaves
4	*Fraxinus griffithii*	Acetone	Oleaceae	Leaves
5		CHCl_3_		Leaves
6		MeOH		Leaves
7	*Phytolacca americana*	Acetone	Phytolaccaceae	Whole plant
8		CHCl_3_		Whole plant
9		MeOH		Whole plant
10	*Elaeocarpus sylvestris*	Acetone	Elaeocarpaceae	Leaves
11		CHCl_3_		Leaves
12		MeOH		Leaves
13		MeOH		Stem
14	*Piper kadsura*	MeOH	Piperaceae	Stem
15	*Itea parviflora*	MeOH	Saxifragaceae	Stem
16	*Callicarpa formosana*	Acetone	Verbenaceae	Leaves
17		MeOH		Leaves
18		MeOH		Stem
19	*Callicarpa kochiana*	Acetone	Verbenaceae	Leaves
20		CHCl_3_		Leaves
21		MeOH		Leaves
22		MeOH		Stem
23		MeOH		Stem
24	*Ficus septica*	Acetone	Moraceae	Leaves
25		CHCl_3_		Leaves
26		MeOH		Leaves
27		MeOH		Fruit
28		MeOH		Heartwood
29	*Ficus sarmentosa* var. *henryi*	Acetone	Moraceae	Leaves
30		CHCl_3_		Leaves
31		MeOH		Leaves
32		MeOH		Stem
33	*Xanthium strumarium*	CHCl_3_	Asteraceae	Fruit
34		MeOH		Fruit
35	*Tribulus terrestris*	Acetone	Zygophyllaceae	Fruit
36		MeOH		Fruit
37	*Cornus officinalis*	Acetone	Cornaceae	Whole plant
38		MeOH		Whole plant
39	*Alisma orientale*	MeOH	Alismataceae	Whole plant
40	*Asparagus cochinchinensis*	MeOH	Asparagaceae	Root
41	*Broussonetia papyrifera*	MeOH	Moraceae	Leaves
42	*Clausena excavata*	MeOH	Rutaceae	Leaves
43	*Cinnamomum insulari-montanum*	MeOH	Lauraceae	Leaves
44	*Lumnitzera racemosa*	MeOH	Combretaceae	Leaves
45	*Pueraria lobata*	MeOH	Fabaceae	Whole plant
46	*Sida acuta*	MeOH	Malvaceae	Whole plant
47	*Scrophularia ningpoensis*	MeOH	Scrophulariaceae	Whole plant
48	*Catharanthus roseus*	MeOH	Apocynaceae	Whole plant above ground

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
