# Peer review of "Xanthium strumarium* Fruit Extract Inhibits ATG4B and Diminishes the Proliferation and Metastatic Characteristics of Colorectal Cancer Cells"

_toxins, 2019, doi:10.3390/toxins11060313_

Round 1

Reviewer 1 Report

The authors present the results of a screen of extracts of a number of medicinal plants for activity against ATG4B, a protein important in autophagy and potentially in cancer progression. The best candidates from the screen underwent further testing to determine their effects on autophagy, cell viability, cell migration, and invasion. The authors found four extracts that significantly inhibited ATG4B. They further found that one of the extracts in particular, XS fruit extract, reduced proliferation, motility, and invasiveness in two lines of colorectal cancer cells. The authors concluded that XS fruit might contain chemicals that are bioactive against colorectal cancer cells via inhibition of ATG4B, which merits further investigation. 

The study was carefully performed with appropriate controls, and the conclusions are well founded based on the data presented. 

Below are some minor considerations that should be modified prior to publication.

Lines 41 and 43: I recommend omitting “Regarding the role of ATG4B in cancer” in both lines. 

Line 69 (Screening plant extracts): The Methods briefly mention the source of the plants, which could be very important. All too frequently, extracts from exotic plants or animals have been found to contain chemicals derived not from the plant or animal itself but from (for example) a common soil bacterium. Extracts from plants from different sources, grown in different environments, may have very different biological activities, so specifics on the source are important. 

I would also like to see more details on the rationale for the choice of plants included in this study as well as the particular plant organs (stem, leaf, etc.) and methods of extraction. The extracts are not comprehensive (i.e. not every method of extraction was used on every organ from every plant), so I would like to know the reasons for including only those from Table 1 in the study. 

Throughout the Results section, I wondered what the effect of the solvent (acetone, CHCl3, MeOH) was on the results? When I read the Methods section, I then understood that the extracts were dried and re-hydrated in medium before being used. You may want to include this detail at the beginning of the Results section just to clarify. 

Line 73: I did not see any “abbreviated names” in Table 1. Is this referring to the number assigned to each extract?

Line 85: In Figure 1C and 1D, the extracts are numbered differently than in Figure 1B. This caused some confusion in the text (line 85), where extracts 33-36 are called 35-38. This is probably because two extracts were blotted twice, accounting for the difference in numbering. This can be remedied by changing the numbering on Figure 1C and 1D to exactly mirror the numbering in Figure 1B. (You may even want to draw a box around the four extracts that were selected for further testing, just to make it obvious to the reader. The numbers on the x-axis are practically too small to read.)

Lines 87-88: “Two of the plant extracts (21 and 26) were randomly tested in different vials and blots to determine if the assay was consistent.” This is an excellent control; however, it appears that they are not particularly consistent—especially for 26-1 and 26-2 in S-Tag blot. This places the overall data here in some doubt, but given that this is a screen of a large number of extracts, and the best candidates underwent validation with a second biochemical test and much further biological testing, this is probably not a significant problem.

Line 99 (Figure 1A and 1B): The substrate protein GABARAPL2 is here called “pro GATE-S” but nowhere else in the paper. This requires explanation. 

Figure 1B: When the concentration of extract increases, the protein level of the enzyme ATG4B decreases. This surprising finding is admitted in the paper but not explained. 

I would call the sample without the enzyme, that gives a maximal signal (retained tags), the “positive control.” And I would call the sample with the enzyme only and no inhibitory extract, that yields minimal signal, the “negative control.” I recommend switching the + and – signs on the graph and in the text. 

Figure 1C and 1D: Explain how data was normalized to allow comparison between immunoblots. 

Line 153 (Figure 3A): Y-axis should be labeled “Luciferase” activity instead of “ATG4B” activity since these are inversely related. 

Line 160: Delete the word “cell” after “harvested.”

Author Response

Manuscript ID: 510168 revised

We thank the referees for a thorough review of our manuscript, and for their helpful suggestions for improvement. We have addressed each point as revised manuscript with track change. We hope the corrections meet with the referees’ satisfaction.

Reviewer 1

Comments and Suggestions for Authors

The authors present the results of a screen of extracts of a number of medicinal plants for activity against ATG4B, a protein important in autophagy and potentially in cancer progression. The best candidates from the screen underwent further testing to determine their effects on autophagy, cell viability, cell migration, and invasion. The authors found four extracts that significantly inhibited ATG4B. They further found that one of the extracts in particular, XS fruit extract, reduced proliferation, motility, and invasiveness in two lines of colorectal cancer cells. The authors concluded that XS fruit might contain chemicals that are bioactive against colorectal cancer cells via inhibition of ATG4B, which merits further investigation. 

The study was carefully performed with appropriate controls, and the conclusions are well founded based on the data presented. 

Below are some minor considerations that should be modified prior to publication.

Lines 41 and 43: I recommend omitting “Regarding the role of ATG4B in cancer” in both lines. 

Ans: Thanks for the suggestion. The statements have been removed from both lines.

Line 69 (Screening plant extracts): The Methods briefly mention the source of the plants, which could be very important. All too frequently, extracts from exotic plants or animals have been found to contain chemicals derived not from the plant or animal itself but from (for example) a common soil bacterium. Extracts from plants from different sources, grown in different environments, may have very different biological activities, so specifics on the source are important. 

Ans: Thanks for the great comments. Plants were collected from Pingtung, southern part of Taiwan. From the 48 active samples, only Phytolacca Americana (No. 7), Asparagus cochinchinensis (No. 40), Sida acuta (No. 46), and Scrophularia ningpoensis (No.47) containing underground part. Plants were then subjected to water rinse cycles until the soil particle could not be observed, and having been classified, dried and chopped very carefully. The contamination from soil bacterium should be eliminated as much as possible. 

I would also like to see more details on the rationale for the choice of plants included in this study as well as the particular plant organs (stem, leaf, etc.) and methods of extraction. The extracts are not comprehensive (i.e. not every method of extraction was used on every organ from every plant), so I would like to know the reasons for including only those from Table 1 in the study. 

Ans: Thanks for the great suggestions. All plants used in this study are typical Formosan plants in southern Taiwan, which are collected and identified by one of the co-authors, Dr. Wei-Yu Lin in Ping Tung County, Taiwan. It is possible that different parts of each plant demonstrated the different anti-ATG4B activity which led us to collect the different parts. Also, comparing the different parts of the same plant will offer the information of the highest anti- ATG4B activity among these parts. It is the same reason in which we used the different kinds of solvents for the extraction. These plants were ground, extracted with the indicated solvent at room temperature for one week. After 3 times of extraction, the extracts were concentrated under vacuum for two weeks to remove the solvent. Thus, the aim of this study was to screen different Formosan plants extracts for potential inhibitors against ATG4B and/or autophagy as a starting point for cancer therapy.

Throughout the Results section, I wondered what the effect of the solvent (acetone, CHCl3, MeOH) was on the results? When I read the Methods section, I then understood that the extracts were dried and re-hydrated in medium before being used. You may want to include this detail at the beginning of the Results section just to clarify. 

Ans: Thanks for the suggestion. These plants were ground, extracted with the indicated solvent at room temperature for one week. After 3 times of extraction, the extracts were concentrated under vacuum for two weeks to remove the solvent.  Thus, the solvents have been minimized to reduce bias from the solvent. The statements have been added in Results section of revised manuscript.

Line 73: I did not see any “abbreviated names” in Table 1. Is this referring to the number assigned to each extract?

Ans: Thanks for pointing out the mistake. We have used numbers to replace abbreviated names in the Table 1 to avoid tight labeling in Figure 1. Thus, we have removed “abbreviated names” from text in line 81 of revised manuscript.

Line 85: In Figure 1C and 1D, the extracts are numbered differently than in Figure 1B. This caused some confusion in the text (line 85), where extracts 33-36 are called 35-38. This is probably because two extracts were blotted twice, accounting for the difference in numbering. This can be remedied by changing the numbering on Figure 1C and 1D to exactly mirror the numbering in Figure 1B. (You may even want to draw a box around the four extracts that were selected for further testing, just to make it obvious to the reader. The numbers on the x-axis are practically too small to read.)

Ans: Indeed, we got the numbered extracts to examine their effects on ATG4B activity without knowing the details of extracts in the beginning.  Then we realized two of them were repeated in the assay.  The number has been corrected and we have marked four potential extracts (No. 33-36) in revised Figure 1C and D. The numbers in axis have been adjusted to bigger size as suggested.

Lines 87-88: “Two of the plant extracts (21 and 26) were randomly tested in different vials and blots to determine if the assay was consistent.” This is an excellent control; however, it appears that they are not particularly consistent—especially for 26-1 and 26-2 in S-Tag blot. This places the overall data here in some doubt, but given that this is a screen of a large number of extracts, and the best candidates underwent validation with a second biochemical test and much further biological testing, this is probably not a significant problem.

Ans: Indeed, though the pattern of recombinant ATG4B and C-Myc tag were similar between 26-1 and 26-2 in the blots. The S-tag is not such consistent between 26-1 and 26-2, suggesting some limitation in this assay. That is why we picked the four hits that showed consistent results in both blots of C-Myc tag and S-tag. As the reviewer’s comments, the hits were further validated by two other secondary assays for confirmation.

Line 99 (Figure 1A and 1B): The substrate protein GABARAPL2 is here called “pro GATE-S” but nowhere else in the paper. This requires explanation. 

Ans: The synonym of GABARAPL2 is GATE16. We apologize the inconsistent labeling and have corrected it to GABARAPL2 in revised Figure 1A and 1B

Figure 1B: When the concentration of extract increases, the protein level of the enzyme ATG4B decreases. This surprising finding is admitted in the paper but not explained. 

Ans: a high dose of some plant extracts decreased the recombinant ATG4B level but had no or little effects on GABARAPL2 cleavage, likely due to aggregation or degradation of ATG4B caused by high dose of the compounds. The statements have been added in line 96-97 of revised manuscript.

I would call the sample without the enzyme, that gives a maximal signal (retained tags), the “positive control.” And I would call the sample with the enzyme only and no inhibitory extract, that yields minimal signal, the “negative control.” I recommend switching the + and – signs on the graph and in the text. 

Ans: We completely agree with comments. We have corrected the labeling of controls in revised Figure 1 as suggested.

Figure 1C and 1D: Explain how data was normalized to allow comparison between immunoblots. 

Ans: The assay only contains purified ATG4B and GABARAPL2 reporter proteins. ATG4B is supposed to be an internal control to normalize. However, several plant extracts affected amount of ATG4B in blots as mentioned above. Thus, the protein levels of S-tag and full-length C-myc-tagged GABARAPL2 were quantitated by ImageJ and normalized by positive control (+) as 100 % inhibition of ATG4B in each blot. The statements have been included in line 118-119 of revised manuscript.

Line 153 (Figure 3A): Y-axis should be labeled “Luciferase” activity instead of “ATG4B” activity since these are inversely related. 

Ans: Thanks for pointing out the mistake. The labeling has been corrected in Figure 3A of revised manuscript.

Line 160: Delete the word “cell” after “harvested.”

Ans: The word ”cell” has been deleted from line 174 or revised manuscript. Thanks.

Reviewer 2 Report

This study demonstrates that Xanthium strumarium fruit extractinhibits ATG4B and thus blocks the autophagic flux. As a consequence, this extract shows anticancer effects in colorectal cancer cells, being able to decrease cell viability above all upon cell starvation. The results are interesting and clearly described. However, in order to transform this submission into a truly convincing study additional work is needed:

1.   It is not clear the rationale behind the choice of the 23 plants, whose extracts were initially screened as potential ATG4B inhibitors. The reason of this specific selection should be explained.

2.   The quality of the Fig. 1 panels C and D should be improved. Axis labels are not visible.

3.   Figure 2 shows that the tested extracts (XS-(fr)-M, XS-(fr)-C, TT-(fr)-A and TT-(fr)) are equally effective at both 1 microg/ml and 10 microg/ml as ATG4B inhibitors using the LC3B-PLA2 biochemical assay. It would be better to display a dose-dependent effect to exclude not specific activities. Above all considering that three out these four extracts are able to inhibit ATG4B only at 100 microg/ml when tested on cells (Fig. 3).

4.   MTT assay should be performed using different extract concentrations, not just two concentrations, in order to obtain a dose response curve and calculate an IC50 value for each of the selected erb extracts.

This information is crucial to choose subapoptotic concentrations to perform wound healing assays and invasion assays, otherwise the results about anti-metastatic effects of the selected extracts might be misinterpreted.

            5.  The would healing results shown in Figure 5 are not very convincing. The        quality of the photos should be improved. 

Author Response

Manuscript ID: 510168 revised

We thank the referees for a thorough review of our manuscript, and for their helpful suggestions for improvement. We have addressed each point as revised manuscript with track change. We hope the corrections meet with the referees’ satisfaction.

Reviewer 2

Comments and Suggestions for Authors

This study demonstrates that Xanthium strumarium fruit extractinhibits ATG4B and thus blocks the autophagic flux. As a consequence, this extract shows anticancer effects in colorectal cancer cells, being able to decrease cell viability above all upon cell starvation. The results are interesting and clearly described. However, in order to transform this submission into a truly convincing study additional work is needed:

1.      It is not clear the rationale behind the choice of the 23 plants, whose extracts were initially screened as potential ATG4B inhibitors. The reason of this specific selection should be explained.

Ans: Thanks for the great suggestions. All plants used in this study are typical Formosan plants in southern Taiwan, which are collected and identified by one of the co-authors, Dr. Wei-Yu Lin in Ping Tung County, Taiwan. It is possible that different parts of each plant demonstrated the different anti-ATG4B activity which led us to collect the different parts. Also, comparing the different parts of the same plant will offer the information of the highest anti- ATG4B activity among these parts. It is the same reason in which we used the different kinds of solvents for the extraction. These plants were ground, extracted with the indicated solvent at room temperature for one week. After 3 times of extraction, the extracts were concentrated under vacuum for two weeks to remove the solvent. Thus, the aim of this study was to screen different Formosan plants extracts for potential inhibitors against ATG4B and/or autophagy as a starting point for cancer therapy. The statements have been added in the first paragraph of Results section in revised manuscript.

2.      The quality of the Fig. 1 panels C and D should be improved. Axis labels are not visible.

Ans: Thanks for the suggestion. We have relabeled the axis and make numbers bigger to be visible. The four potential extracts (No. 33-36) were also marked with red line in revised Figure 1C and 1D.

3.      Figure 2 shows that the tested extracts (XS-(fr)-M, XS-(fr)-C, TT-(fr)-A and TT-(fr)) are equally effective at both 1 microg/ml and 10 microg/ml as ATG4B inhibitors using the LC3B-PLA2 biochemical assay. It would be better to display a dose-dependent effect to exclude not specific activities. Above all considering that three out these four extracts are able to inhibit ATG4B only at 100 microg/ml when tested on cells (Fig. 3).

Ans: Yes, we actually tried various dose of plant extracts for LC3B-PLA2 assay (below). The plant extracts inhibited fluorescent signal in cellular ATG4B assay in nanomolar range (7.8 nM). However, the dosage could be varied between in vitro and cell-based assays, depending on sensitivity and stability of reporter assays, such as cell penetration ability, and plasma stability of compounds. Thus, the concentration of plant extracts used in vitro assay and in cellular assay was different.

4.      MTT assay should be performed using different extract concentrations, not just two concentrations, in order to obtain a dose response curve and calculate an IC50 value for each of the selected erb extracts.

This information is crucial to choose subapoptotic concentrations to perform wound healing assays and invasion assays, otherwise the results about anti-metastatic effects of the selected extracts might be misinterpreted.

Ans: Thanks for the comments. We titrated different doses of plant extracts and measured cell viability with Celltiter Glo assay using colon normal epithelia cells FHC as a control. As shown in Fig. S1, the IC50 of XS-(fr)-C or TT-(fr)-M were >100 ug/ml, which also reduce viability of the colon normal epithelia cells. Thus, we are not interested in higher concentration of these plant extracts in cancer cell proliferation due to no selective effects. 

Regarding the effects of extracts on cell migration and invasion, since 10 or 50 mg/ml of XS-(fr)-C or TT-(fr)-M had no or little effects on cell viability (Fig. S1), human colorectal cancer HCT116 and DLD-1 cells were treated with 10 or 50 mg/ml XS-(fr)-C or TT-(fr)-M for the cell migration and invasion assays. Thus, XS-(fr)-C could inhibit cancer cell migration and invasion, not because of altering cell viability. On the other hand, compared with two-dimensional culture, three-dimensional tumor cell culture is relatively more capable of reproducing complicated microenvironments in vivo, such as low nutrients and oxygen in central part of tumors, which induces autophagy. Therefore, to precisely inspect the effects of plant extracts in cancer cells, HCT116 cells were cultured for sphere formation and treated with XS-(fr)-C or TT-(fr)-M (Fig. 6E and 6F). We found XS-(fr)-C and TT-(fr)-M significantly increased dead cells in tumor spheres. These statements haven added in last paragraph of Results section on line 160-164 of revised manuscript.

            5.  The would healing results shown in Figure 5 are not very convincing. The        quality of the photos should be improved. 

Ans: Thanks for the suggestion. We have adjusted the images to improve the resolution of Figure 5.

Round 2

Reviewer 2 Report

In my opinion, the revised version of the manuscript "Xanthium strumarium fruit extract inhibits ATG4B and diminishes the proliferation and metastatic characteristics of colorectal cancer cells" deserves to be published on Toxins.